# The Antioxidant Potential of the Mediterranean Diet as a Predictor of Weight Loss after a Very Low-Calorie Ketogenic Diet (VLCKD) in Women with Overweight and Obesity

**DOI:** 10.3390/antiox12010018

**Published:** 2022-12-22

**Authors:** Ludovica Verde, Maria Dalamaga, Xavier Capó, Giuseppe Annunziata, Maria Hassapidou, Annamaria Docimo, Silvia Savastano, Annamaria Colao, Giovanna Muscogiuri, Luigi Barrea

**Affiliations:** 1Centro Italiano Per la Cura e il Benessere del Paziente con Obesità (C.I.B.O), Dipartimento di Medicina Clinica e Chirurgia, Unit of Endocrinology, Federico II University Medical School of Naples, Via Sergio Pansini 5, 80131 Napoli, Italy; 2Dipartimento di Medicina Clinica e Chirurgia, Unit of Endocrinology, Federico II University Medical School of Naples, Via Sergio Pansini 5, 80131 Napoli, Italy; 3Department of Biological Chemistry, Medical School, National and Kapodistrian University of Athens, 11527 Athens, Greece; 4Translational Research In Aging and Longevity (TRIAL) Group, Health Research Institute of the Balearic Islands (IdISBa), 07120 Palma, Spain; 5Department of Pharmacy, University of Naples Federico II, Via Domenico Montesano 59, 80131 Naples, Italy; 6Department of Nutritional Sciences and Dietetics, International Hellenic University, 57400 Thessaloniki, Greece; 7Cattedra Unesco “Educazione alla Salute e Allo Sviluppo Sostenibile”, University Federico II, 80131 Napoli, Italy; 8Dipartimento di Scienze Umanistiche, Università Telematica Pegaso, Via Porzio, Centro Direzionale, Isola F2, 80143 Napoli, Italy

**Keywords:** antioxidant, Mediterranean diet, weight loss, very low-calorie ketogenic diet (VLCKD), obesity

## Abstract

Obesity involves a chronic state of low-grade inflammation, which is linked to the development of several comorbidities. Recently, the very low-calorie ketogenic diet (VLCKD) has gained great interest in the treatment of obesity, almost ousting the ancient and healthy Mediterranean diet (MD). However, because these dietary regimens exploit different pathophysiological mechanisms, we hypothesize that adherence to the MD may play a role in determining the efficacy of the VLCKD. We enrolled 318 women (age 38.84 ± 14.37 years; BMI 35.75 ± 5.18 kg/m²) and assessed their anthropometric parameters, body compositions, and adherence to the MD (with the PREvención con DIetaMEDiterránea (PREDIMED) questionnaire) at baseline. The anthropometric parameters and body composition were repeated at the end of the VLCKD. At the end of the VLCKD, the women with high adherence to the MD achieved the best results in terms of weight loss and improved body composition. Specifically, the women who were above the median of fat mass (FM)% reduction had the best MD pattern, characterized by a higher consumption of extra virgin olive oil (EVOO), fruits, vegetables, and red wine, as well as a higher adherence to the MD than the women who were below the same median. In a multiple regression analysis, the PREDIMED score was the main predictor of the FM% reduction score and came in first, followed by fruit, EVOO, and glasses of wine, in predicting the percentage reduction in FM. A PREDIMED score value of > 5 could serve as a threshold to identify patients who are more likely to lose FM at the end of the VLCKD. In conclusion, high adherence to the MD resulted in higher VLCKD efficacy. This could be due to the antioxidant and anti-inflammatory properties of the MD, which are capable of establishing a metabolic set-up that is favorable to the onset of more effective ketosis.

## 1. Introduction

Obesity, defined as a body mass index (BMI) of ≥ 30 kg/m^2^ [1], is an expanding problem with an underestimated pathology. The most recent World Health Organization (WHO) report refers to an increase in the incidence of obesity, which unfortunately has not been defeated yet [2]. In fact, 59% of European adults and almost one in three children (29% of males and 27% of females) are overweight or suffer from obesity [2]. In particular, up to 2016, the age-standardized prevalence of overweight and obesity among adults in Italy was 58.5% and 19.9%, respectively [3]. Interestingly, in parallel with obesity, a state of inflammation occurs that is mediated by the overproduction of reactive oxygen species (ROSs) [4]. In addition, it is worth remembering that obesity is accompanied by numerous comorbidities and chronic diseases, such as type 2 diabetes mellitus (T2DM), dyslipidemia, hypertension, polycystic ovary syndrome, various neoplasms, and sleep apnea syndrome, among others [5]. Weight loss, although not easy to achieve, is a fundamental goal that is achieved by reducing energy intake and increasing energy expenditure [6]. There are several possible strategies for weight loss, among which are anti-obesity drugs, bariatric surgery, and a variety of dietary interventions [7,8]. New anti-obesity drugs with very promising prospects are, however, limited by their non-negligible cost, some contraindications, and possible side effects that do not make them accessible to all patients with obesity [9]. Bariatric surgery has well-known evidence of efficacy for weight loss and remission of T2DM and metabolic syndrome but results in numerous complications that make it only suitable for individuals with severe obesity and no contraindications to the procedure [10]. Among the various nutritional approaches to treating obesity, the one that is most widely used and has the most extensive supporting scientific evidence is the Mediterranean diet (MD), which is a healthy dietary pattern that is characterized by a wide variety of plant foods [11]. The MD includes foods such as extra virgin olive oil (EVOO), vegetables, fruits, legumes, nuts, red wine, and whole-grain cereals. It is characterized by less saturated fat, more mono- and polyunsaturated fat, and a high intake of bioactive compounds, including polyphenols and omega-3 fatty acids with anti-inflammatory and antioxidant potency. These anti-inflammatory and antioxidant properties, together with certain intrinsic characteristics (a high-carbohydrate/high-fat scheme with a high ratio of monounsaturated to saturated fats, good fiber intake, and others), also make the MD an appropriate nutritional approach for weight loss [11]. The beneficial effects of the MD can be attributed to its different foods that are rich in anti-inflammatory and antioxidant properties [12]. However, despite the well-known beneficial effects of the MD on human health, obesity issues persist and do not stop their growth. In this context, individuals with obesity who have failed to achieve weight loss with macronutrient-balanced nutritional approaches can benefit from an already well-known and promising nutritional approach, which is the very low-calorie ketogenic diet (VLCKD) [13,14]. This diet consists of a marked daily restriction of carbohydrates (usually less than 30g/day), together with a relative increase in the amount of protein (about 43% of the total energy) and fat (about 44% of the total energy). The total daily energy intake remains very low, around 800 kcal/day. The latest evidence reports an impressive efficacy of the VLCKD in the treatment of obesity, dyslipidemia, and, in general, for most cardiovascular risk factors [15,16]. In addition to the low calorie intake, reduced insulin levels, increased glucagon levels, and, especially, the production of ketone bodies, which exert various additional beneficial effects, weight loss remains more rapid than with other diets [13,14]. Excellent adherence, a common problem with all dietary interventions, is also favored by the perception of an unfamiliar and temporary dietary pattern, as well as by the anorectic effect of ketone bodies [17]. Finally, the relative maintenance of protein mass compared to starvation is also an advantage [18]. However, although several studies have highlighted the efficacy of the VLCKD in treating obesity, not all individuals who undertake the VLCKD succeed in achieving the same result in terms of weight loss and changes in body composition. However, the VLCKD could also have a spectrum of efficacy, which is mostly due to the efficacy of reaching ketosis. Since both the MD and the VLCKD exploit different pathophysiological mechanisms, we hypothesize that previous adherence to the MD may play a role in determining the efficacy of the active phase of the VLCKD.

Thus, the aim of this study was to evaluate the differences in weight loss and body composition improvements based on adherence to the MD prior to nutritional intervention in a cohort of women with overweight and obesity undergoing the VLCKD.

## 2. Materials and Methods

### 2.1. Design and Setting

We enrolled 318 women with overweight or obesity who were referred clinically for weight loss at the Centro Italiano per la cura e il Benessere del paziente con Obesità (C.I.B.O), Division of Endocrinology, Department of Clinical Medicine and Surgery, University Federico II in Naples, Italy. The population was selected based on the following criteria:a)Inclusion criteria: 18 years of age or older; overweight or obesity in accordance with a BMI of ≥ 27.0 kg/m^2^; naïve subjects (no previous treatment with anti-obesity drugs or bariatric surgery);b)Exclusion criteria: type 1 diabetes mellitus or T2DM on insulin therapy or latent autoimmune diabetes in adults; renal, hepatic, cardiac (NYHA III-IV), or respiratory insufficiency; unstable angina; stroke or myocardial infarction in the last 12 months; cardiac arrhythmias; psychiatric and eating disorders; alcohol and drug addiction; active and/or severe infections; rare metabolic diseases; pregnancy; lactation.

### 2.2. Anthropometric Measurements and Physical Activity

Anthropometric measurements were performed at the beginning (5 days before the start of the VLCKD) and at the end of the active phase of the VLCKD (after 45 days) by the same qualified nutritionist. The body weight was determined with a calibrated beam scale (0.1 kg approximation, Seca 711; Seca, Hamburg, Germany), and the height was determined with a wall stadiometer (0.5 cm approximation, Seca 711; Seca, Hamburg, Germany). BMI is a commonly used quantitative measure of adiposity [19]. Thus, the weight and height were combined in the following formula:Weight (kg)/Height (m^2^)

The result was then used to categorize the BMI (overweight, obesity I, II, and III classes) according to WHO criteria. Finally, the waist circumference (WC) was measured with a non-elastic tape measure at the natural indentation or halfway between the lower edge of the rib cage and the iliac crest if no natural indentation was visible (0.1 cm approximation), as reported by the National Center for Health Statistics [20]. Thirty minutes of physical activity per day was the threshold for defining physically active women, as previously reported [21].

### 2.3. Bioelectrical Impedance Analysis

A bioelectrical impedance analysis (BIA) was performed at the beginning and at the end of the active phase of the VLCKD by the same qualified nutritionist to assess body composition. The assessment was performed using a phase-sensitive BIA system with an 800-µA current at a frequency of 50 kHz (BIA 101, Akern Bioresearch, Florence, Italy), according to the recommendations of the European Society of Parental and Enteral Nutrition [22]. The phase angle (PhA) is calculated as the relationship between the resistance (R) of tissues, which depends mainly on tissue hydration, and their reactance (Xc), which is related to cellularity, cell size, and cell membrane integrity, according to the following formula:PhA (°, degrees) = Xc/R* (180/π),
as reported in previous evidence [23]. The same operator and instrument performed the BIA determinations under strictly standardized conditions to avoid inter-observer and inter-instrument variability.

### 2.4. Assessment of the Adherence to MD

Prior to the start of the VLCKD, a qualified nutritionist assessed the adherence to the MD using the PREvención con DIetaMEDiterránea (PREDIMED) questionnaire, which consists of 14 items that investigate the consumption of EVOO, fruit, vegetables, nuts, legumes, red meat, poultry, fish, animal fats, sugary drinks, sweets, and sofrito [24]. As already widely reported [25], the PREDIMED score calculation provided the following MD adherence categories: 0–5, low adherence; 6–9, average adherence; ≥10, high adherence [24].

### 2.5. VLCKD Intervention

Women who fulfilled the inclusion criteria were administered the VLCKD protocol with a meal replacement, consisting of active, re-education, and maintenance phases. For the VLCKD, a commercial weight-loss program (New Penta, Cuneo, Italy) was used. After a nutritional status assessment, a qualified nutritionist prepared the active-phase diet, which was prescribed by an endocrinologist, as already reported [26]. The protocol was carried out according to the European Association for the Study of Obesity (EASO) guidelines [14]. As stated in the guidelines, the VLCKD was characterized as follows: a daily energy intake of approximately <800 kcal with 13% of carbohydrates (<30 g/day), 43% of protein (1.2–1.5 g/kg of ideal body weight), and 44% of lipids (including 10 g/day of EVOO as the only fat seasoning). This composition was achieved by using high biological value protein preparations (formulated on the basis of peas, eggs, soy, and whey), containing between 15 and 18 g of high biological value protein. Four or five meals a day were provided, according to the individuals’ needs. Guidance was also given on the vegetables to be consumed with regard to type, quantity (unlimited or grammed), and method of preparation/cooking. No other food was allowed beyond the meal replacements and the proposed vegetables. In addition, each woman was recommended to be well hydrated daily by consuming only water. Given the high caloric restrictions, all of the women were prescribed vitamin and mineral supplements to maintain the physiological acid/base balance (PentaCal, Penta, s.r.l., Cuneo, Italy) (B-complex vitamins, vitamins C and E, and minerals, including potassium, sodium, magnesium, calcium, and omega-3 fatty acids), in accordance with international recommendations [14]. A qualified nutritionist ascertained compliance with the VLCKD and physical activity recommendations through individual telephone counselling and also instructed the women to measure their β-hydroxybutyrate from capillary blood using test strips once a week (Optium Xceed Blood Glucose and Ketone Monitoring System; Abbott Laboratories, Chicago, IL, USA) in the morning on an empty stomach, preferably at the same time.

### 2.6. Statistical Analysis

The Kolmogorov–Smirnov test was used to test the data distributions. The continuous variables were expressed as mean ± standard deviation (SD) when normally distributed. The categorical variables were expressed as numbers and percentages (%). The variations were analyzed using a paired *t*-test for the normally distributed variables. A chi-square (χ^2^) test was used to test for differences in the frequency distributions. The differences among the categories of adherence to the MD were analyzed by a between-groups ANOVA test followed by a Bonferroni *post-hoc* test. A Pearson’s correlation was used to assess the association between the PREDIMED score at baseline, weight loss, and changes in body composition pre/post-intervention. A partial correlation was performed to adjust the associations for the confounding factors (BMI, fat mass (FM), and WC). A multinomial logistic regression model, odds ratio (OR), *p*-value, 95% interval confidence (IC), and R^2^ were performed to assess the associations among the Δ% FM kg with single items of the PREDIMED questionnaire and adherence to the MD categories. In addition, two multiple linear regression analysis models (the stepwise method), expressed as R^2^, beta (*β*), and *t*, with the PREDIMED score as a dependent variable, were used to estimate the predictive value of the weight and body composition parameters (Model 1), and with the Δ% FM kg as a dependent variable, were used to estimate the predictive value of the PREDIMED score and single items of the PREDIMED questionnaire (Model 2). A receiver operator characteristic (ROC) curve analysis was performed to determine the area under the curve (AUC), criterion, sensitivity, specificity, standard error, 95% IC, and the cut-off values for the PREDIMED score in detecting patients who were most likely to lose FM at the end of the active phase of the VLCKD. A statistical analysis was performed according to standard methods using the statistical package for the social sciences software 26.0 (SPSS/PC; SPSS, Chicago, IL, USA). A *p* value of < 0.05 was considered statistically significant. Since this was a pilot study, no power calculations were performed. Therefore, all of the findings need to be confirmed by larger clinical trials.

## 3. Results

A total of 318 women with overweight and obesity (BMI 35.75 ± 5.18 kg/m^2^), aged 38.84 ± 14.37 years, met the inclusion/exclusion criteria and were included in this study. All of the women were evaluated at baseline and at the end of the active phase of the VLCKD. Once a week, a nutritionist conducted a telephone interview to confirm compliance with the VLCKD, ketosis status, and any changes in the physical activity levels of the women.

Before the start of the VLCKD, the PREDIMED score was 6.38 ± 2.36. Most of the women (172, 54.2%) had average adherence to the MD, followed by low (117, 36.8%) and high adherences to the MD (29, 9.1%); Figure 1.

Compared to the baseline, no women changed their physical activity levels during the 45 days of the VLCKD (101, 31.8% vs. 101, 31.8%; χ^2^ = 0.01, *p* = 0.932). Table 1 reports the anthropometric measurements and body composition parameters of the study population at baseline and at the end of the active phase of the VLCKD, respectively. At the end of the active phase of the VLCKD, the mean percentage weight loss was −7.35 ± 2.72%, with a statistically significant decrease in WC (*p* < 0.001). Of interest, the distribution of the women across the BMI categories was significantly modified, with an increase in the prevalence of women with normal weight (+2.2%), overweight (+15.4%), and grade I obesity (+0.3%), while the prevalence of women with grade II and grade III obesity dropped to −5.9% and −12.0%, respectively. As for the body composition parameters, all of the parameters statistically changed at the end of the VLCKD. The FM (kg and %) was statistically reduced (*p* < 0.001), and the fat-free mass (FFM) (kg and %) (*p* < 0.001) and PhA (*p* < 0.001) were statistically increased.

Table 2 reports the anthropometric measurements and body composition parameters of the study population according to adherence to the MD. At the baseline, the subgroups of women did not differ in age (*p* = 0.264), R (*p* = 0.291), FFM (kg) (*p* = 0.136), and PhA (*p* = 0.091). The women with high adherence to the MD had significantly lower BMIs (*p* < 0.001), WCs (*p* < 0.001), and FMs (kg and %) (*p* < 0.001) than the women with low adherence to the MD. At the end of the active phase of the VLCKD, the women with high adherence to the MD had lost more body weight (*p* < 0.001), WC (*p* < 0.001), and FM (kg and %) (*p* < 0.001) and had increased FFM (%) (*p* < 0.001) and PhA (*p* = 0.010) compared to the women with low adherence to the MD; Table 2.

Table 3 reports the physical activity levels and BMI categories of the study population according to adherence to the MD. Both at baseline and at the end of the active phase of the VLCKD, the women had not changed their physical activity levels (*p* = 0.059). At baseline, the lowest percentage of women had grade III obesity (*p* < 0.001). At the end of the active phase of the VLCKD, all of the BMI categories changed, showing a reduction in grade III obesity (*p* < 0.001) and an increase in normal-weight women (*p* < 0.001).

Table 4 shows the response frequency of the single items included in the PREDIMED questionnaire of the study population. EVOO, which was the main culinary lipid used, was the main food of the MD consumed by the study population (84.3%), followed by low consumption of red/processed meats (51.3%).

The single items included in the PREDIMED questionnaire and the categories of adherence to the MD were examined by stratifying the study population above and below the median of the Δ% FM kg (−14.84%); Table 5. The women that were above the median of the Δ% FM kg had the best Mediterranean dietary pattern, characterized by the highest consumption of EVOO (*p* < 0.001), fruit (*p* < 0.001), vegetables (*p* < 0.001), and red wine (*p* < 0.001), and high adherence to the MD (*p* = 0.002) compared to the women that were below the median of the Δ% FM kg.

Figure 2 shows the PREDIMED scores above and below the median of the Δ% FM kg (−14.84%). The women that were above the median of the Δ% FM kg had the highest PREDIMED score than the women that were below the median of the Δ% FM kg (*p* < 0.001).

Table 6 reports the simple and after-adjustment correlations for the confounding variables among the PREDIMED scores and changes in the anthropometric measurements and body composition parameters at the end of the active phase of the VLCKD. The PREDIMED score correlates with all of the parameters except for the Δ% WC (*p* = 0.093) and Δ% PhA (*p* = 0.430).

The results of the bivariate proportional odds ratio (OR) model, performed to assess the association of the Δ% FM kg with the quantitative variables of the PREDIMED questionnaire and adherence to the MD categories, are reported in Table 7. The highest values of the Δ% FM kg were significantly associated with the consumption of all foods of the MD included in the PREDIMED questionnaire, except for the intake of poultry more than red meats (*p* = 0.276) and the use of sofrito sauce ≥ 2/week (*p* = 0.195). Of interest, the highest values of the Δ% FM kg were also significantly associated with all categories of adherence to the MD (*p* < 0.001 for all).

To compare the relative predictive power of the PREDIMED score associated with the weight loss and body composition changes at the end of the active phase of the VLCKD, we performed a multiple linear regression analysis model. In this model, the Δ% FM kg represents the most predictive parameter of adherence to the MD, which was entered in the first step (*p* < 0.001). The results are reported in Table 8.

To compare the relative predictive power of the PREDIMED score and the single items of the PREDIMED questionnaire associated with the Δ% FM kg, we performed a second multiple linear regression analysis. In this second model, the PREDIMED score was entered in the first step (*p* < 0.001), followed by fruits ≥ 3 servings/day (*p* = 0.010), the use of EVOO as the main culinary lipid (*p* = 0.001), and wine glasses ≥ 7/week (*p* = 0.002); Table 9.

A ROC analysis was performed to determine the cut-off value of the PREDIMED score, which was predictive of the greatest FM loss at the end of the active phase of the VLCKD (above the median value of the Δ% FM kg, −14.84%), as shown in Figure 3. A value of the PREDIMED score of >5 (*p* < 0.001; AUC 0.825; standard error 0.024; 95% IC 0.778 to 0.872) could serve as a threshold for identifying patients who are most likely to lose FM at the end of the active phase of the VLCKD.

## 4. Discussion

The aim of this prospective observational study was to determine whether adherence to the MD could predict the efficacy of the active phase of the VLCKD in terms of weight loss and body composition changes in women with obesity.

Only a small percentage of the women had high adherence to the MD (9.1%), and more than 80% of the women reported consuming EVOO as their main condiment at the expense of using other, less healthy condiments. As expected, the women with high adherence to the MD had the best anthropometric and body composition parameters. In fact, there is consistent scientific evidence that adherence to the MD is inversely associated with adiposity, in particular, abdominal adiposity [27,28].

Our study revalidated the efficacy of the VLCKD in the treatment of obesity by allowing the enrolled women to achieve significant weight and FM losses and an increase in PhA after 45 days of the active phase. PhA is a measure derived from BIA analyses and, being an indicator of cellular health and integrity [29], is, therefore, a valuable tool for determining the inflammatory state associated with many diseases, such as obesity [30]. Thus, along with the reduction in anthropometric parameters and improvements in body composition, the VLCKD also shows promising effects in reducing the low-grade inflammation associated with obesity. In this regard, in a pilot study of 260 women with obesity, we recently reported changes in PhA during the active phase of the VLCKD that occurred very early and independently of weight loss and were negatively associated with high-sensitive C-reactive protein levels [26]. The efficacy of the VLCKD on anthropometric parameters, body composition, and, more recently, on inflammatory status is mainly mediated by ketone bodies (acetone, acetoacetate, and β-hydroxybutyrate) [31,32,33], which are produced at the mitochondrial level of different tissues, mainly from the oxidation of fatty acids, and are exported to peripheral tissues to be used as an energy source [34]. These new and interesting effects of the VLCKD are underpinned by several mechanisms exerted by ketone bodies, including activation of the peroxisome proliferator-activated receptor gamma and hydroxycarboxylic acid receptor 2, inhibition of NF-κB activation, and inhibition of the NLRP3 inflammasome [35].

As novel findings, we report that the women who had higher adherence to the MD at baseline showed greater weight and FM losses with a greater PhA increase. The high adherence to the MD was statistically associated with percentage decreases in BMI, WC, and FM, even after adjustments for the main confounding factors, including BMI, WC, Xc, FM, and PhA, at baseline. Interestingly, at the end of the active phase of the VLCKD, 13.8% of the women with high adherence to the MD became normal weight, while none of the women with low adherence to the MD achieved the same results, with no difference in their physical activity levels. Analyzing the individual foods from the PREDIMED questionnaire, the participants above the median value of FM loss showed a higher consumption of typical Mediterranean pattern foods, such as EVOO, vegetables, fruit, red wine, fish, nuts, and white meat instead of red meat, and a lower consumption of butter, soda drinks, commercial sweets, and confectionery. These foods were also statistically associated with FM loss. In fact, the participants with the highest PREDIMED scores were located above the median FM loss value.

Studies have shown that obesity causes mitochondrial dysfunction, or the inability of mitochondria to produce and maintain sufficient levels of adenosine triphosphate (ATP), which is the result of an imbalance in nutrient signal input, energy production, and oxidative respiration [36]. Indeed, low-grade inflammation and excessive nutrient intake (as is often the case in obesity) can lead to an increased production of ROSs and cause oxidative stress. Oxidative stress, in turn, can trigger mitochondrial changes, called mitochondrial dysfunction [36]. Mitochondria are essential for cellular energy metabolism, as they generate ATP by oxidizing carbohydrates, lipids, and proteins [37]. Of note, the production of ketone bodies during the VLCKD occurs at the hepatic mitochondrial level [37], which allows us to speculate that weight excess could negatively affect these processes as well.

Recently, the so-called “mitochondrial nutrients”, which include polyphenols, plant-derived compounds, and polyunsaturated fatty acids (PUFAs), have attracted the attention of researchers with the idea that improving mitochondrial structure and function is a plausible strategy to correct mitochondrial dysfunction [38]. In this respect, scientific evidence has demonstrated the beneficial effects of various bioactive MD compounds on mitochondrial dysfunction [38]. Such valuable effects might explain why the high adherence to the MD and the parallel consumption of certain specific nutrients might enhance the efficacy of the VLCKD.

In our study, EVOO, the key condiment in the MD, was associated with better outcomes in terms of weight loss and improvements in body composition following the VLCKD. Hydroxytyrosol, a polyphenol from olive oil, has been shown to be effective in regulating mitochondrial dysfunction in animal models by exploiting its ability to modulate the mitochondrial apoptotic pathway in skeletal muscle and the liver [39]. Hydroxytyrosol can also enhance mitochondrial biogenesis (specifically by increasing mtDNA and the number of mitochondria) by increasing the expression of the genes involved (PGC-1α, NRF-1, and TFAM) via the AMPK pathway [40].

Similar results were seen for some compounds found in abundance in red wine and plant products, such as fruit and vegetables, which we also found to be associated with improved efficacy of the VLCKD. In this regard, a study in mice reported that resveratrol, which is found in foods such as grapes and berries, improved mitochondrial activity by competing with NAD+ in a solubilized complex [41]. Additionally, in mice, resveratrol appears to be able to prevent metabolic diseases (such as insulin resistance and obesity) by improving mitochondrial function through the activation of genes such as PGC-1α and SIRT-1 [42]; this result was later corroborated in another very similar study [43]. Among the antioxidants, lycopene, which belongs to the phytochemical family of carotenes and is abundant in foods such as tomatoes and grapefruit, also improved mitochondrial dysfunction by exerting an anti-inflammatory effect on mice [44]. In particular, lycopene regulated the expression of certain genes, such as SIRT1, PGC1α, Cox5b, Cox7a1, Cox8b, and Cycs, as well as many respiratory chain complexes [44]. Finally, ellagic acid, which is found in strawberries and walnuts, prevented metabolic disorders by acting on mitochondria in the following two ways: directly, by decreasing the amount of ROSs and mitochondrial damage, and indirectly, by restoring the total dehydrogenase activity in the mitochondria [45].

We cannot omit to mention PUFAs, which are precursors of many other substances in the body (particularly some of these involved in blood pressure regulation and inflammatory responses) and fundamental components of cell membranes [46]. It is now known that excessive amounts of n-6 PUFAs (and a high n-6/n-3 ratio), as found in Western diets, promote the pathogenesis of many diseases, including cardiovascular diseases, cancer, and inflammatory and autoimmune diseases, while higher levels of n-3 PUFAs (and a low n-6/n-3 ratio) exert suppressive effects. Interestingly, it has been observed that a diet with a high n-3/n-6 ratio, as is the case with MD, shows an enhancement of the activities of mitochondrial complexes, accompanied by an attenuation of fumaric acid and oxidative stress [46]. In particular, walnuts, which are part of the MD and are the only tree nuts to provide n-3 PUFAs (with the n-6/n-3 ratio being around 4/1), have antioxidant and anti-inflammatory activities via the activation of the Nrf2/ARE and down-regulation of the NF-κB pathways [47].

However, we showed that the main predictor of efficacy of the VLCKD was the PREDIMED score, i.e., the whole Mediterranean nutritional model. In particular, a cut-off value above 5 was the best predictor of FM loss after the active phase of the VLCKD. This result is of fundamental importance, more than the individual foods, since it is the entire dietary pattern that probably sets up the body in a more favorable condition for weight loss and improved body composition through the synergy of the different nutrients. In addition to the individual mechanisms discussed above, which are limited to animal models, this can be partly explained by the intake of polyphenols and monounsaturated fatty acids in the Mediterranean dietary pattern, which, in addition to improving mitochondrial dysfunction, have been shown to inhibit de novo lipogenesis, improve peripheral insulin sensitivity [48], activate β-oxidation, and increase lipolysis [49]. It is probably also for this reason that the VLCKD in subjects with high adherence to the MD finds a more supportive ground for implementing its weight loss mechanisms.

In brief, it is possible to hypothesize that high adherence to the MD may, in part, attenuate mitochondrial dysfunction in women with obesity, resulting in a more efficient production of ketone bodies, and, in part, support those metabolic pathways directed toward fat oxidation, ultimately resulting in a basal metabolic set-up that is more conducive to weight loss.

We are aware that there are some limitations in our study, but also several strengths. First, the cross-sectional nature of this study did not allow for any statements on a causal relationship between adherence to the MD and weight loss and body composition improvements associated with the active phase of the VLCKD. Second, because we did not assess KB levels in the blood, we could not assess any differences in KB production. However, ketur capillary tests, which were performed independently by our patients, confirmed the ketosis status of the study population. Third, the PREDIMED score, although easy to perform by the participants, only allows for relative but not absolute statements regarding the degree of adherence to the MD. Nevertheless, the PREDIMED questionnaire was performed face-to-face and was not self-reported in order to minimize any bias related to filling out the questionnaire. Of interest, the PREDIMED questionnaire allows one to provide feedback to the subjects immediately after the interview is completed. In addition, to avoid inter-operator variability, only one expert nutritionist evaluated the anthropometric measures, administered the PREDIMED questionnaire, and assessed, executed, and interpreted the BIA measurements. Moreover, we increased the homogeneity of the patients’ samples by including only women with overweight or obesity in order to improve the power of the study. Additionally, we did not perform a quantitative assessment of inflammation markers and oxidative stress; however, we have already widely reported that PhA may be a marker of inflammation and is associated with the MD [26,30]. In addition, with regard to logistic regression analyses, when the incidence of events/number of patients is high (as in this study), then the odds ratio can be misleading as it exaggerates the size of the effects [50]. Finally, although the gold standard for body composition assessments is dual energy X-ray absorptiometry (DXA), we used BIA in our study. However, BIA is also a validated and widely used method for assessing body compartments in subjects with obesity, with close concordance with DXA results [51,52].

## 5. Conclusions

In conclusion, this is the first study to report that the efficacy of the VLCKD is greater in women with high adherence to the MD before starting the ketogenic diet. This could be due to the presence of interesting bioactive compounds in the Mediterranean diet, which together and in the context of the MD could lead to a favorable metabolic set-up for the onset of a more effective ketosis. Furthermore, as a possible translational application, these results suggest that a specific cut-off value for the PREDIMED score (<5) can help to identify women in whom the VLCKD would achieve lower efficacy results. It is, therefore, possible to hypothesize that these women could boost the efficacy of the VLCKD with an earlier, brief nutritional intervention aimed at improving adherence to the MD, particularly by increasing foods such as EVOO and fruits before starting the ketogenic protocol.

## Figures and Tables

**Figure 1 antioxidants-12-00018-f001:**
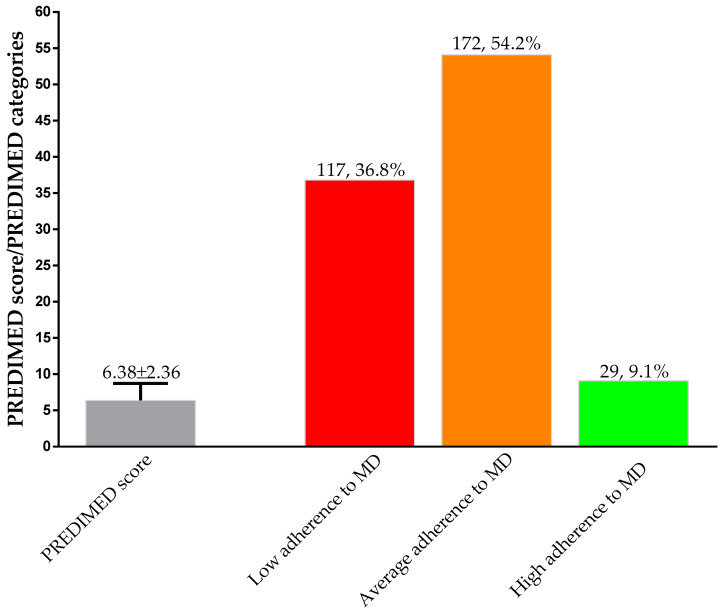
Adherence to the MD in the study population at baseline. PREDIMED, PREvención con DIetaMEDiterránea; MD, Mediterranean diet.

**Figure 2 antioxidants-12-00018-f002:**
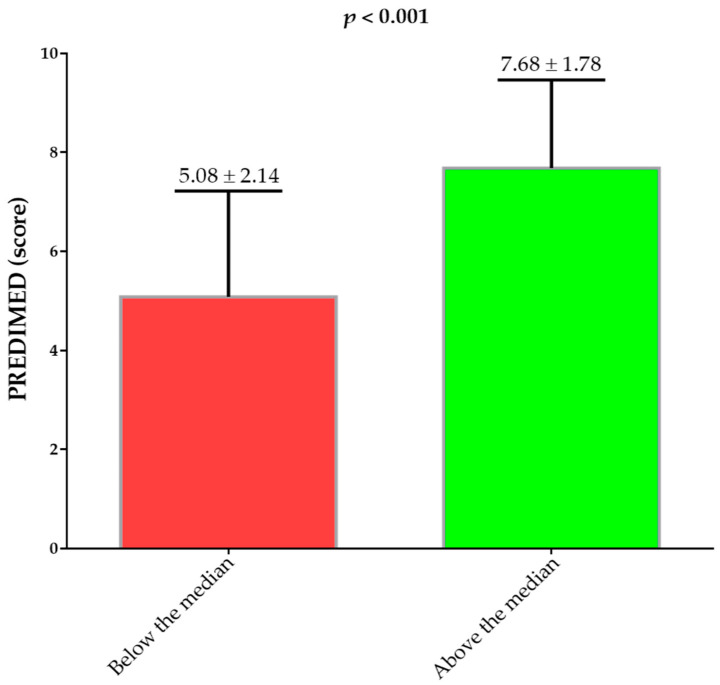
PREDIMED scores of the study population above and below the median of the Δ% FM kg. PREDIMED, PREvención con DIetaMEDiterránea. A *p*-value in bold denotes a significant difference (*p* < 0.05).

**Figure 3 antioxidants-12-00018-f003:**
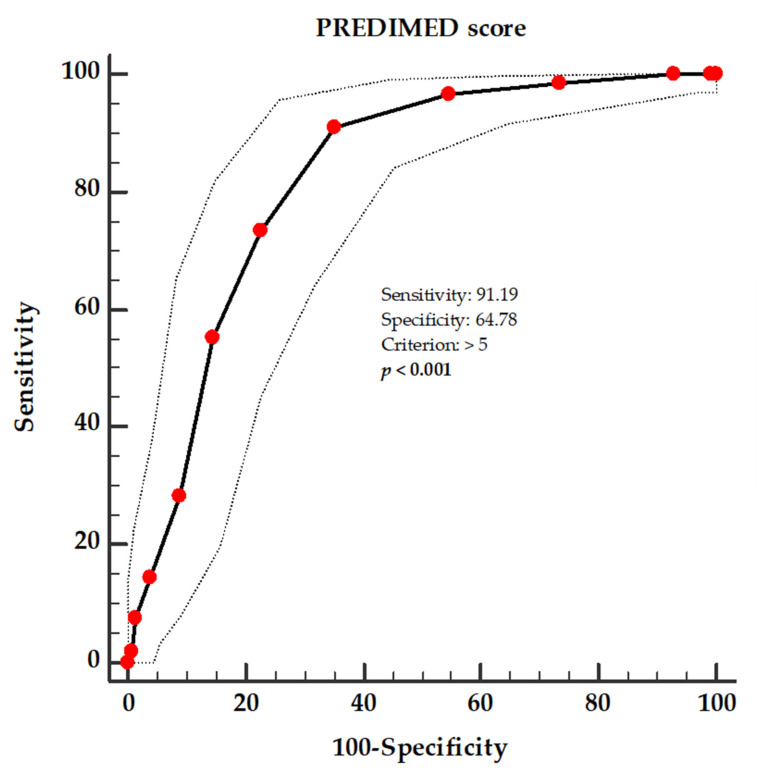
ROC for the predictive values of the PREDIMED score in detecting patients who are most likely to lose FM at the end of the active phase of the VLCKD. A *p*-value in bold denotes a significant difference (*p* < 0.05).

**Table 1 antioxidants-12-00018-t001:** Anthropometric measurements and body composition parameters of the study population at baseline and at the end of the active phase of the VLCKD.

Parameters N = 318	Baseline	End of Active Phase of VLCKD	∆%	** p*-Value
Weight (kg)	95.15 ± 15.89	88.14 ± 14.91	−7.35 ± 2.72	**<0.001**
BMI (kg/m^2^)	35.75 ± 5.18	33.11 ± 4.88	−7.36 ± 2.73	**<0.001**
Normal weight (n, %)	-	7, 2.2	2.2	χ^2^ = 5.20, ***p* = 0.023**
Overweight (n, %)	47, 14.8	96, 30.2	15.4	χ^2^ = 20.79, ***p* < 0.001**
Grade I obesity (n, %)	104, 32.7	105, 33.0	0.3	χ^2^ = 0.00, *p* = 1.000
Grade II obesity (n, %)	100, 31.4	81, 25.5	−5.9	χ^2^ = 2.50, *p* = 0.114
Grade III obesity (n, %)	67, 21.1	29, 9.1	−12.0	χ^2^ = 16.80, ***p* < 0.001**
WC (cm)	104.84 ± 14.58	98.39 ± 13.63	−5.96 ± 4.56	**<0.001**
BIA				
R (Ω)	477.36 ± 69.94	483.81 ± 67.29	1.76 ± 8.35	**0.003**
Xc (Ω)	47.64 ± 9.72	51.17 ± 9.52	8.75 ± 15.18	**<0.001**
FM (kg)	40.98 ± 12.64	34.83 ± 11.52	−15.06 ± 8.85	**<0.001**
FM (%)	42.24 ± 6.59	38.66 ± 6.94	−8.53 ± 7.78	**<0.001**
FFM (kg)	54.16 ± 5.79	53.24 ± 6.02	−1.65 ± 4.76	**<0.001**
FFM (%)	57.76 ± 6.59	61.33 ± 6.94	6.33 ± 5.60	**<0.001**
PhA (°)	5.71 ± 0.86	6.04 ± 0.81	6.78 ± 11.77	**<0.001**

Data are expressed as mean  ±  standard deviation (SD) or as numbers (n) and percentages (%). VLCKD, very low-calorie ketogenic diet; Δ%, percentage change; BMI, body mass index; WC, waist circumference; BIA, bioelectrical impedance analysis; R, resistance; Xc, reactance; FM, fat mass; FFM, fat-free mass; PhA, phase angle. A *p*-value in bold denotes a significant difference (*p* < 0.05).

**Table 2 antioxidants-12-00018-t002:** Anthropometric measurements and body composition parameters of the study population according to the categories of adherence to the MD at baseline and at the end of the active phase of the VLCKD.

Parameters N = 318	Low Adherence to MD 117, 36.8%	Average Adherence to MD 172, 54.1%	High Adherence to MD 29, 9.1%	*p-*Value
Baseline parameters				
Age (years)	39.46 ± 14.41	37.84 ± 14.05	42.24 ± 15.88	0.264
Weight (kg)	100.90 ± 16.97 ^a,b^	93.28 ± 14.19 ^a^	83.06 ± 11.22	**<0.001**
BMI (kg/m^2^)	38.13 ± 5.04 ^a,b^	34.90 ± 4.71 ^a^	31.23 ± 3.66	**<0.001**
WC (cm)	110.68 ± 13.98 ^a,b^	102.29 ± 13.89	95.51 ± 12.26	**<0.001**
R (Ω)	469.29 ± 64.06	481.87 ± 76.10	483.21 ± 50.88	0.291
Xc (Ω)	46.01 ± 9.48	48.26 ± 10.19	50.51 ± 6.45	**0.038**
FM (kg)	46.00 ± 13.65 ^a,b^	39.36 ± 11.00 ^a^	30.32 ± 7.62	**<0.001**
FM (%)	44.81 ± 6.16 ^a,b^	41.52 ± 6.26 ^a^	36.10 ± 50.08	**<0.001**
FFM (kg)	54.89 ± 5.83	53.90 ± 5.79	52.73 ± 5.47	0.136
FFM (%)	55.19 ± 6.16 ^a,b^	58.48 ± 6.26 ^a^	63.90 ± 5.08	**<0.001**
PhA (°)	5.60 ± 0.88	5.73 ± 0.86	5.97 ± 0.61	0.091
Parameters at the end of active phase of VLCKD				
Weight (kg)	94.65 ± 15.58 ^a,b^	85.81 ± 13.03 ^a^	75.76 ± 10.30	**<0.001**
Δ% weight	−6.89 ± 2.76	−7.60 ± 2.71	−7.75 ± 2.41	0.065
BMI (kg/m^2^)	35.78 ± 4.61 ^a,b^	32.09 ± 4.27 ^a^	28.49 ± 3.46	**<0.001**
Δ% BMI	−6.11 ± 2.30 ^a,b^	−7.99 ± 2.57	−8.72 ± 3.34	**<0.001**
WC (cm)	104.22 ± 12.79 ^a,b^	95.79 ± 12.99	90.24 ± 12.02	**<0.001**
Δ% WC	−5.61 ± 4.26	−6.17 ± 4.53	−6.33 ± 5.96	0.539
R (Ω)	488.74 ± 66.64	481.41 ± 69.66	478.13 ± 55.38	0.592
Δ% R	4.54 ± 9.58 ^a,b^	0.34 ± 7.37	−1.02 ± 4.99	**<0.001**
Xc (Ω)	50.25 ± 10.33	51.71 ± 9.39	51.66 ± 6.35	0.428
Δ% Xc	10.41 ± 16.98	8.57 ± 13.91	3.15 ± 13.75	0.068
FM (kg)	41.06 ± 11.85 ^a,b^	32.53 ± 9.46 ^a^	23.33 ± 6.63	**<0.001**
Δ% FM	−10.29 ± 6.73 ^a,b^	−17.07 ± 8.14	−23.17 ± 10.15	**<0.001**
FM (%)	42.70 ± 5.80 ^a,b^	37.28 ± 6.05 ^a^	30.51 ± 5.82	**<0.001**
Δ% FM	−4.49 ± 6.06 ^a,b^	−10.06 ± 7.23 ^a^	−15.78 ± 8.76	**<0.001**
FFM (kg)	53.59 ± 6.00	53.29 ± 5.93	51.47 ± 6.54	0.236
Δ% FFM	−2.34 ± 4.61 ^b^	−1.07 ± 4.07	−2.24 ± 7.98	0.065
FFM (%)	57.30 ± 5.80 ^a,b^	62.69 ± 6.07 ^a^	69.49 ± 5.82	**<0.001**
Δ% FFM	−4.49 ± 6.06 ^a,b^	−10.06 ± 7.23	−15.78 ± 8.76	**<0.001**
PhA (°)	5.86 ± 0.86 ^b^	6.15 ± 0.81	6.17 ± 0.50	**0.010**
Δ% PhA	5.50 ± 11.67	8.24 ± 11.77	4.08 ± 11.58	0.063

Data are expressed as mean  ±  standard deviation (SD). MD, Mediterranean diet; Δ%, percentage change; BMI, body mass index; WC, waist circumferences; R, resistance; Xc, reactance; FM, fat mass; FFM, fat-free mass; PhA, phase angle; VLCKD, very low-calorie ketogenic diet. A *p*-value in bold denotes a significant difference (*p* < 0.05). ^a^ *p* < 0.05 vs. high adherence to the MD (at least significant difference post-hoc analysis for multiple comparisons). ^b^ *p* < 0.05 vs. average adherence to the MD (at least significant difference post-hoc analysis for multiple comparisons).

**Table 3 antioxidants-12-00018-t003:** Physical activity levels and BMI categories of the study population according to the categories of adherence to the MD at baseline and at the end of the active phase of the VLCKD.

Parameters N = 318	Low Adherence to MD 117, 36.8%	Average Adherence to MD 172, 54.1%	High Adherence to MD 29, 9.1%	*p-*Value
Baseline parameters				
Physical activity				
Yes (n, %)	35, 29.9	52, 30.2	15, 51.7	χ^2^ = 5.56 *p* = 0.059
No (n, %)	82, 70.1	120, 69.8	14, 18.3	
BMI categories				
Normal weight (n, %)	0, 0	0, 0	0, 0	-
Overweight (n, %)	1, 0.9	32, 18.6	14, 48.3	χ^2^ = 45.84, ***p* < 0.001**
Grade I obesity (n, %)	37, 31.6	57, 33.1	10, 34.5	χ^2^ = 0.12, *p* = 0.943
Grade II obesity (n, %)	36, 30.8	60, 34.9	4, 13.8	χ^2^ = 5.16, *p* = 0.076
Grade III obesity (n, %)	43, 36.8	23, 13.4	1, 3.4	χ^2^ = 28.85, ***p* < 0.001**
Parameters at the end of active phase of VLCKD				
Physical activity				
Yes (n, %)	35, 29.9	52, 30.2	15, 51.7	χ^2^ = 5.56 *p* = 0.059
No (n, %)	82, 70.1	120, 69.8	14, 18.3	
BMI categories				
Normal weight (n, %)	0, 0%	3, 1.7	4, 13.8	χ^2^ = 20.90, ***p* < 0.001**
Overweight (n, %)	16, 13.7	61, 35.5	19, 65.5	χ^2^ = 34.59, ***p* < 0.001**
Grade I obesity (n, %)	37, 31.6	65, 37.8	3, 10.3	χ^2^ = 8.62, ***p* = 0.014**
Grade II obesity (n, %)	42, 35.9	36, 20.9	3, 10.3	χ^2^ = 12.06, ***p* = 0.002**
Grade III obesity (n, %)	22, 18.8	7, 4.1	0, 0	χ^2^ = 21.44, ***p* < 0.001**

Data are expressed as numbers (n) and percentages (%). MD, Mediterranean diet; BMI, body mass index; VLCKD, very low-calorie ketogenic diet. A *p*-value in bold denotes a significant difference (*p* < 0.05).

**Table 4 antioxidants-12-00018-t004:** Response frequency of dietary components included in the PREDIMED Questionnaire of the study population.

Questions of PREDIMED questionnaire N = 318		
	n	%
Use of EVOO as main culinary lipid	268	84.3
EVOO > 4 tablespoons	176	55.3
Vegetables ≥ 2 servings/day	139	43.7
Fruits ≥ 3 servings/day	128	40.3
Red/processed meats < 1/day	163	51.3
Butter, cream, margarine < 1/day	150	47.2
Soda drinks < 1/day	149	46.9
Wine glasses ≥ 7/week	64	20.1
Legumes ≥ 3/week	142	44.7
Fish/seafood ≥ 3/week	109	34.3
Commercial sweets and confectionery ≤ 2/week	118	37.1
Tree nuts ≥ 3/week	126	39.6
Poultry more than red meats	145	45.6
Use of sofrito sauce ≥ 2/week	151	47.5

Data are expressed as numbers (n) and percentages (%). PREDIMED, PREvención con DIetaMEDiterránea; EVOO, extra virgin olive oil.

**Table 5 antioxidants-12-00018-t005:** The single items included in the PREDIMED questionnaire and the categories of adherence to the MD of the study population above and below the median of the Δ% FM kg.

Questions of PREDIMED Questionnaire n = 318	Below the Median n = 159		Above the Median n = 159			
	n	%	n	%	χ^2^	** p*-Value
Use of EVOO oil as main culinary lipid	115	72.3	153	96.2	32.49	**<0.001**
EVOO > 4 tablespoons	79	49.7	97	61.0	3.68	0.055
Vegetables ≥ 2 servings/day	46	28.9	93	58.5	27.04	**<0.001**
Fruits ≥ 3 servings/day	31	19.5	97	61.0	55.24	**<0.001**
Red/processed meats < 1/day	66	41.5	97	61.0	11.34	**0.001**
Butter, cream, margarine < 1/day	62	39.0	88	55.3	7.89	**0.005**
Soda drinks < 1/day	64	40.3	85	53.5	5.05	**0.025**
Wine glasses ≥ 7/week	16	10.1	48	30.2	18.80	**<0.001**
Legumes ≥ 3/week	63	39.6	79	49.7	2.86	0.091
Fish/seafood ≥ 3/week	43	27.0	66	41.5	6.76	**0.009**
Commercial sweets and confectionery ≤ 2/week	42	26.4	76	47.8	14.67	**0.001**
Tree nuts ≥ 3/week	54	34.0	72	45.3	3.80	**0.049**
Poultry more than red meats	60	37.7	85	53.5	7.30	**0.007**
Use of sofrito sauce ≥ 2/week	66	41.5	85	53.5	4.09	0.053
Low adherence to MD	103	64.8	14	8.8	104.7	**<0.001**
Average adherence to MD	50	31.4	122	76.7	63.84	**<0.001**
High adherence to MD	6	3.8	23	14.5	9.71	**0.002**

Data are expressed as numbers (n) and percentages (%). PREDIMED, PREvención con DIetaMEDiterránea; EVOO, extra virgin olive oil; MD, Mediterranean diet. A *p*-value in bold denotes a significant difference (*p* < 0.05).

**Table 6 antioxidants-12-00018-t006:** Simple and after-adjustment correlations for the confounding variables among the PREDIMED score and changes in the anthropometric measurements and body composition parameters at the end of the active phase of the VLCKD.

Parameters	PREDIMED Score N = 318			
	Simple Correlation		After Adjusted for Confounding Variables	
	*r*	*p-*Value	*r*	*p-*Value
Δ% weight	−0.149	**0.008**	−0.154	**0.006**
Δ% BMI	−0.377	**<0.001**	−0.409	**<0.001**
Δ% WC	−0.094	0.093	−0.139	**0.014**
Δ% R	−0.349	**<0.001**	−0.380	**<0.001**
Δ% Xc	−0.179	**0.001**	−0.107	**0.050**
Δ% FM	−0.556	**<0.001**	−0.551	**<0.001**
Δ% FFM	0.188	**0.001**	0.528	**<0.001**
Δ% PhA	0.044	0.430	0.182	**0.001**

PREDIMED, PREvención con DIetaMEDiterránea; Δ%, percentage change; BMI, body mass index; WC, waist circumferences; R, resistance; Xc, reactance; FM, fat mass; FFM, fat-free mass; PhA, phase angle. A *p*-value in bold denotes a significant difference (*p* < 0.05).

**Table 7 antioxidants-12-00018-t007:** Bivariate proportional odds ratio model that was used to assess the relationship between the Δ% FM kg and the PREDIMED questionnaire and adherence to the MD categories.

	Δ% FM (n = 318)			
Questions	OR	*p-*Value	95% IC	R^2^
Use of EVOO as main culinary lipid	0.85	**<0.001**	0.81–0.89	0.160
EVOO > 4 tablespoons	0.96	**0.002**	0.94–0.99	0.030
Vegetables ≥ 2 servings/day	0.92	**<0.001**	0.89–0.95	0.112
Fruits ≥ 3 servings/day	0.91	**<0.001**	0.88–0.94	0.120
Red/processed meats < 1/day	0.96	**0.004**	0.94–0.99	0.026
Butter, cream, margarine < 1/day	0.97	**0.024**	0.94–1.00	0.016
Soda drinks < 1/day	0.96	**0.003**	0.93–0.98	0.029
Wine glasses ≥ 7/week	0.92	**<0.001**	0.89–0.95	0.074
Legumes ≥ 3/week	0.97	**0.012**	0.94–0.99	0.020
Fish/seafood ≥ 3/week	0.94	**<0.001**	0.91–0.97	0.059
Commercial sweets and confectionery ≤ 2/week	0.95	**<0.001**	0.93–0.98	0.044
Tree nuts ≥ 3/week	0.98	**0.050**	0.95–1.00	0.011
Poultry more than red meats	0.99	0.276	0.96–1.01	0.004
Use of sofrito sauce ≥ 2/week	0.98	0.195	0.96–1.00	0.005
Low adherence to MD	0.88	**<0.001**	0.85–0.91	0.181
Average adherence to MD	0.94	**<0.001**	0.92–0.97	0.057
High adherence to MD	0.89	**<0.001**	0.85–0.94	0.078

FM, fat mass; PREDIMED, PREvención con DIetaMEDiterránea; EVOO, extra virgin olive oil; MD, Mediterranean diet; OR, odds ratio; IC, interval confidence. A *p*-value in bold denotes a significant difference (p < 0.05).

**Table 8 antioxidants-12-00018-t008:** Multiple regression analysis model (the stepwise method) with the PREDIMED score as a dependent variable to estimate the predictive value of changes in weight and body composition parameters at the end of the active phase of the VLCKD.

Parameters	Multiple Regression Analysis n = 318			
	R^2^	*β*	*t*	* *p-*Value
Δ% FM	0.307	−0.556	−11.91	**<0.001**

FM, fat mass. A *p*-value in bold denotes a significant difference (*p* < 0.05).

**Table 9 antioxidants-12-00018-t009:** Multiple regression analysis model (the stepwise method) with the Δ% FM kg at the end of the active phase of the VLCKD as a dependent variable to estimate the predictive value of the PREDIMED score and single items of the PREDIMED questionnaire.

Parameters	Multiple Regression analysis N = 318			
	R^2^	*β*	*t*	* *p-*Value
PREDIMED score	0.307	−0.556	−11.91	**<0.001**
Fruits ≥ 3 servings/day	0.325	−0.129	−2.60	**0.010**
Use of EVOO as main culinary lipid	0.339	−0.177	−3.37	**0.001**
Wine glasses ≥ 7/week	0.356	−0.147	−3.05	**0.002**

PREDIMED, PREvención con DIetaMEDiterránea; EVOO, extra virgin olive oil. A *p*-value in bold denotes a significant difference (*p* < 0.05).

## Data Availability

The data presented in this study are available on request from the corresponding author. The data are not publicly available due to ethical restrictions.

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
