# Peer review of "The Antioxidant Potential of the Mediterranean Diet as a Predictor of Weight Loss after a Very Low-Calorie Ketogenic Diet (VLCKD) in Women with Overweight and Obesity"

_antioxidants, 2022, doi:10.3390/antiox12010018_

Round 1

Reviewer 1 Report

1. Authors need to work on Abstract.

2. Define obesity as the first sentence in the introduction with the following manuscript (https://www.sciencedirect.com/science/article/pii/S1319562X19301639)

3. Define the prevalence of obesity in SPAIN

4.  Aim was not clear.

5. Does there was any criteria for recruiting 318 women.

6. Title does not reflect as only women will be recruited in this study?

7. Why did authors recruit only Women in this study?

8.  Lane:125, add the following citation for BMI (https://www.ncbi.nlm.nih.gov/pmc/articles/PMC8017326/)

9. Authors need to expand in detail about sub-section 2.5

10. Tables and stats in the results section was found to be fine.

11. Current study results should compare with documented results in the global studies.

Author Response

Reviewer #1

  1. Authors need to work on Abstract.

R:  We thank the reviewer for the comment. We have edited the abstract.

  1. Define obesity as the first sentence in the introduction with the following manuscript

(https://www.sciencedirect.com/science/article/pii/S1319562X19301639)

R: We thank the reviewer for the suggestion. We have better defined obesity according to the suggested reference (line 51).

  1. Define the prevalence of obesity in SPAIN

R: We thank the reviewer for the suggestion. However, the study was conducted in Italy, so we thought it more appropriate to add up-to-date Italian prevalence data on obesity and overweight (lines 55-58).

  1. Aim was not clear.

R: We thank the reviewer for the observation. We have better specified the purpose of the study (lines 104-107).

  1. Does there was any criteria for recruiting 318 women.

R: As this was a pilot study, no power calculations were performed. Thus, we enrolled in the study all women who came to our outpatient clinic with the aim of losing weight, who underwent VLCKD and who met the inclusion and exclusion criteria (section 2.1 of methods) for this study.

  1. Title does not reflect as only women will be recruited in this study?

R: We thank the reviewer for the comment and have amended the title.

  1. Why did authors recruit only Women in this study?

R: There is a strong disproportion in the subjects attending our outpatient clinic, with a higher prevalence of female subjects. For this reason, we increased the homogeneity of the patient sample by including only women with overweight or obesity, in order to improve the power of the study.

  1. Line 125, add the following citation for BMI

(https://www.ncbi.nlm.nih.gov/pmc/articles/PMC8017326/)

R: We thank the reviewer for the suggestion. We have better defined BMI according to the suggested reference (lines 129-130)

  1. Authors need to expand in detail about sub-section 2.5

R: We thank the reviewer for the comment. We have added some notions regarding the VLCKD protocol used (lines 172-177).

  1. Tables and stats in the results section was found to be fine.

R: We sincerely thank the reviewer for the positive comment.

  1. Current study results should compare with documented results in the global studies.

R: To our knowledge, no studies with similar results are yet available in the literature. In fact, we have not found any other studies that have evaluated the efficacy of a dietary intervention based on the previous adherence to MD of enrolled subjects.

Reviewer 2 Report

This study revealed that women following a very low-calorie ketogenic diet with previous adherence to Mediterranean Diet reached the best results in terms of weight loss and body composition improvements. The topic is important. However, the following suggestions should be attended to: 

Recent published studies dealing with the efficacy and safety of very low calorie ketogenic diet in overweight and obese patients should also be cited: doi: 10.3390/nu13061804doi: 10.1007/s11154-019-09514-y

Lines 55-56: sentence is not in English

Line 138: “Parenteral”!

Line 143: as “evidence”, reference #21 should be sufficient here, the other two could be viewed as self-citations

Line 152: the same observation as above; please choose one reference from 24, 25, 26

Line 159: same here. I suggest “... as already reported [29].

Line 161: “characterized

Line 182: “analyzed

Lines 432-441: it should be added here that “walnuts, part of MD and the only tree nuts to provide n-3 PUFAs, with the n- 6/n-3 ratibeing around 4/1, have antioxidant and anti-inflammatory activities via the activation of Nrf2/ARE and down-regulation of NF-kB pathways (doi: 10.3390/antiox11071412)

Lines 459-477: another limitation should be mentioned - “when the incidence of events/number of patients is high (as in this study), then odds ratio can be misleading as it exaggerates the size of the effect (doi: 10.1503/cmaj.101715)”. 

In your study, the average BMI was 35.75 ± 5.18 kg/m2; please discuss the statement “BIA has been shown to be valid with BMIs to 34 kg/m2 (doi:10.1016/j.clnu.2004.09.012)”

Line 477 (same as Lines 143, 152, 159): too many self-citations (ref. 21 and 29 should be adequate). Eventually, very recent references can be added here: doi: 10.1155/2022/7165126doi: 10.3389/fendo.2022.924199

All References should be rechecked and updated: #33 and #52 are the same; many are incomplete (1, 2, 3, 7, 8, 10, 14, 32, 39, 51, 53)

Author Response

Reviewer #2

  1. Recent published studies dealing with the efficacy and safety of very low-calorie ketogenic diet in overweight and obese patients should also be cited: doi: 10.3390/nu13061804; doi: 10.1007/s11154-019-09514-y

R: We thank the reviewer for the suggestion. We have added the suggested references (lines 92).

  1. Lines 55-56: sentence is not in English; Line 138: “Parentereal”!
  2. Line 161: “characterized”
  3. Line 182: “analyzed”

R: We thank the reviewer for the observation. We have corrected the typos.

  1. Line 143: as “evidence”, references #21 should be sufficient here, the other two could be viewed as self-citations:
  2. Line 159: same here. I suggest ”…as already reported [29].”
  3. Line 477 (same as lines 143, 152, 159): too any self-citations (ref 21 and 29 should be adequate). Eventually, very recent references can be added here: doi: 10.1155/2022/7165126; doi: 103389/fendo.2022.924199

R: We thank the reviewer. We have corrected according to the suggestions.

  1. Lines 432-441: it should be added here that “walnuts, part of MD and the only tree nuts to provide n3 PUFAs, with the n-6/n-3 ratio being around 4/1, have antioxidant and anti-inflammatory activities via the activation of Nrf2/ARE and down-regulation of NF-kB pathways (doi: 10.3390/antiox11071412)

R: We thank the reviewer for the useful suggestion. We have added what was suggested (lines 458-460).

  1. Lines 459-477: another limitation should be mentioned “when the incidence of events/number of patients is high (as in this study), then odds ratio can be misleading as it exaggerates the size of the effects (doi:10.1503/cmaj.101715)

R: We thank the reviewer for the correct observation and have added the suggested limit pf the study (lines 496-499).

  1. In your study, the average BMI was 35.75 ± 5.18 kg/m2; please discuss the statement “BIA has been shown to be valid with BMIs to 34 kg/m2 (doi:10.1016/j.clnu.2004.09.012)

R: We thank the reviewer for the observation. We understand that currently available data suggest that BIA works well in healthy subjects and in patients with stable water and electrolyte balance with a validated BIA equation appropriate for age, sex and race. Clinical use of BIA in subjects at the extremes of the BMI range or with abnormal hydration cannot be recommended for routine evaluation of patients until further validation has demonstrated the accuracy of the BIA algorithm in such conditions. However, BIA is a simple and reproducible method for assessing changes in body composition also in subjects with obesity, with a close concordance with DXA (gold standard) results. Nevertheless, in accordance with the reviewer's right observation, we have added this aspect to the limitations of the study (lines 499-502).

  1. All References should be rechecked and updated: #33 and #52 are the same: many are incomplete (1, 2, 3, 7, 8, 10, 14, 32, 29, 51, 53)

R: We thank the reviewer. We have corrected the bibliography.

Round 2

Reviewer 1 Report

Authors have justified all the raised quires.